# ED2: An Environment Dynamics Decomposition Framework for World Model Construction

## Abstract

Model-based reinforcement learning methods achieve significant sample efficiency in many tasks, but their performance is often limited by the existence of the model error. To reduce the model error, previous works use a single well-designed network to fit the entire environment dynamics, which treats the environment dynamics as a black box. However, these methods lack to consider the environmental decomposed property that the dynamics may contain multiple sub-dynamics, which can be modeled separately, allowing us to construct the world model more accurately. In this paper, we propose the Environment Dynamics Decomposition (ED2), a novel world model construction framework that models the environment in a decomposing manner. ED2 contains two key components: *sub-dynamics discovery* (SD2) and *dynamics decomposition prediction* (D2P). SD2 discovers the sub-dynamics in an environment and then D2P constructs the decomposed world model following the sub-dynamics. ED2 can be easily combined with existing MBRL algorithms and empirical results show that ED2 significantly reduces the model error and boosts the performance of the state-of-the-art MBRL algorithms on various continuous control tasks. [1]

## 1 Introduction

Reinforcement Learning (RL) is a general learning framework for solving sequential decision-making problems and has made significant progress in many fields (Mnih et al., 2015; Silver et al., 2016; Vinyals et al., 2019; Schrittwieser et al., 2019). In general, RL methods can be divided into two categories regarding whether a world model is constructed for the policy deriving: model-free RL (MFRL) and model-based RL (MBRL). MFRL methods train the policy by directly interacting with the environment, which results in good asymptotic performance but low sample efficiency. By contrast, MBRL methods improve the sample efficiency by modeling the environment, but often with limited asymptotic performance and suffer from the model error (Lai et al., 2020; Kaiser et al., 2020).

Existing MBRL algorithms can be divided into four categories according to the paradigm they follow: the first category focuses on generating imaginary data by the world model and training the policy with these data via MFRL algorithms (Kidambi et al., 2020; Yu et al., 2020); the second category leverages the differentiability of the world model, and generates differentiable trajectories for policy optimization (Deisenroth & Rasmussen, 2011; Levine & Koltun, 2013; Zhu et al., 2020); the third category aims to obtain an accurate value function by generating imaginations for temporal difference (TD) target calculation (Buckman et al., 2018; Feinberg et al., 2018); the last category of works focuses on reducing the computational cost of the policy deriving by combining the optimal control algorithm (e.g. model predictive control) with the learned world models (Chua et al., 2018; Okada & Taniguchi, 2019; Argenson & Dulac-Arnold, 2020). Regardless of paradigms, the performance of all existing MBRL algorithms depends on the accuracy of the world model. The more accurate the world model is, the more reliable data can be generated, and finally, the better policy performance can be achieved. Therefore, improving the world model accuracy is critical in MBRL.

To this end, various techniques have been proposed to improve the model accuracy. For example, rather than directly predict the next state, some works construct a world model for the state change

---

[1] Our code is open source and available at `https://github.com/ED2-source-code/ED2`

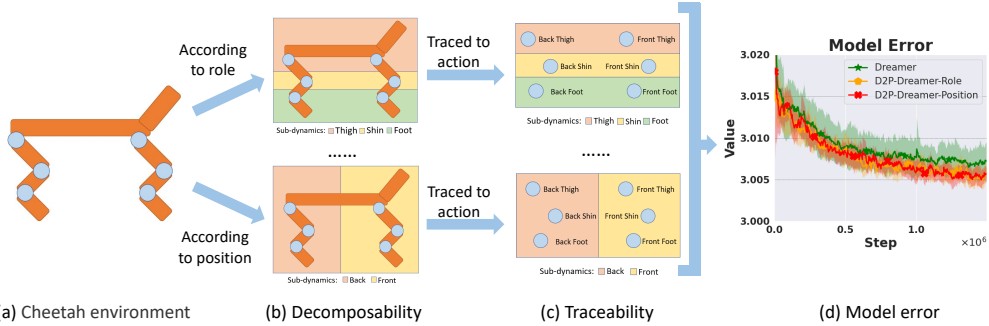

Figure 1: (a) The Cheetah task with six action dimensions. (b) The dynamics can be decomposed into multiple sub-dynamics in various ways, each sub-dynamics is described with different background colors. (c) Dynamics can be traced to the impact caused by the action, and for each sub-dynamics, we show the meanings of the action dimensions it traced to. (d) The model error comparison on the Cheetah task of Dreamer and D2P-Dreamer methods (D2P-Dreamer-Role/Position correspond to decompose the dynamics according to role/position and model each sub-dynamics separately).

prediction (Luo et al., 2019; Kurutach et al., 2018). Model ensemble is also widely used in model construction for uncertainty estimation, which provides a more reliable prediction (Janner et al., 2019; Pan et al., 2020). To reduce the model error in long trajectory generation, optimizing the multi-step prediction errors is also an effective technique (Hafner et al., 2019). However, these techniques improve the environment modeling in a black-box way, which ignores the inner decomposed structure of environment dynamics. For example, Figure 1 (a) shows the Cheetah task from DeepMindControl (DMC) Suite tasks, where the dynamics can be decomposed in various ways. Figure 1 (b) shows various decomposition on the dynamics: according to the role of sub-dynamics, we can decompose it into: $\{thigh, shin, foot\}$; alternatively, according to the position of sub-dynamics, we can decompose it into: $\{back, front\}$. Figure 1 (d) shows that no matter whether we decompose the Cheetah task according to role or position, modeling each decomposed sub-dynamics separately can significantly reduce the model error of the existing MBRL algorithm (e.g. Dreamer (Hafner et al., 2020)).

Inspired by the above example, we propose environment dynamics decomposition (ED2), a novel world model construction framework that models the dynamics in a decomposing fashion. ED2 contains two main components: *sub-dynamics discovery* (SD2) and *dynamics decomposition prediction* (D2P). SD2 is proposed to decompose the dynamics into multiple sub-dynamics, which can be flexibly designed and we also provide three alternative approaches: complete decomposition, human prior, and the clustering-based method. D2P is proposed to construct the world model from the decomposed dynamics in SD2, which models each sub-dynamics separately in an end-to-end training manner. ED2 is orthogonal to existing MBRL algorithms and can be used as a backbone to easily combine with any MBRL algorithm. Experiment shows ED2 improves the model accuracy and boosts the performance significantly when combined with existing MBRL algorithms.

## 2 BACKGROUND

### 2.1 REINFORCEMENT LEARNING

Given an environment, we can define a finite-horizon partially observable Markov decision process (POMDP) as $(S, A, R, P, \gamma, O, \Omega, T)$, where $S \in \mathbb{R}^n$ is the state space, and $A \in \mathbb{R}^m$ is the action space, $R : S \times A \to \mathbb{R}$ denotes the reward function, $P : S \times A \to S$ denotes the environment dynamics, $\gamma$ is the discount factor. The agent receives an observation $o \in \Omega$, which contain partial information about the state $s \in S$. $O$ is the observation function, which mapping states to probability distributions over observations. The decision process length is denoted as $T$.

Let $\eta$ denote the expected return of a policy $\pi$ over the initial state distribution $\rho_0$. The goal of an RL agent is to find the optimal policy $\pi^*$ which maximizes the expected return:

$$\pi^* = \arg\max_{\pi} \eta[\pi] = \arg\max_{\pi} \mathbb{E}_{\pi}[\sum_{t=0}^{T} \gamma^t R(s_t, a_t)],$$

where $s_0 \sim \rho_0, o_t \sim O(\cdot|s_t), a_t \sim \pi(\cdot|o_t), s_{t+1} \sim P(\cdot|s_t, a_t)$. If the environment is fully observable, i.e., $\Omega = S$ and $O$ is an identity function, POMDP is equivalent to the MDP: $(S, A, R, P, \gamma, T)$.

## 2.2 REPRESENTATIVE WORLD MODELS IN MBRL

The world model is a key component of MBRL that directly impacts policy training. World models are usually formulated with latent dynamics (Janner et al., 2019; Hafner et al., 2020), and the general form of the latent dynamics model can be summarized as follows:

$$\begin{aligned} \text{Latent transition kernel:} \quad & h_t = f(s_{\leq t-1}, a_{\leq t-1}) \\ \text{Stochastic state function:} \quad & p(s_t|h_t) \\ \text{Reward function:} \quad & p(r_t|h_t) \end{aligned}$$

The latent transition kernel (shorthand as kernel) predicts the latent state $h_t$ with input $s_{\leq t-1}$ and $a_{\leq t-1}$. Based on latent state $h_t$, the stochastic state function and reward function decode the state $s_t$ and reward $r_t$. For the partially observable environment, two additional functions are required:

$$\begin{aligned} \text{Observation function:} \quad & p(o_t|s_t) \\ \text{Representation function:} \quad & p(s_t|h_t, o_t) \end{aligned}$$

In general, world models mainly differ at the implementation of kernel, which can be roughly divided into two categories: **with non-recurrent kernel** and **with recurrent kernel**. The formal definition of both kernels are as follows:

$$h_t = \begin{cases} f(s_{t-1}, a_{t-1}) & \text{With non-recurrent kernel} \\ f(h_{t-1}, s_{t-1}, a_{t-1}) & \text{With recurrent kernel} \end{cases}$$

Non-recurrent kernel are relatively basic kernel for modeling, which are often implemented as Fully-Connected Networks. Non-recurrent kernel takes the current state $s_{t-1}$ and action $a_{t-1}$ as input, outputs the latent state prediction $h_t$. Compare to non-recurrent kernel, recurrent kernel is implemented as RNN and takes the additional input $h_{t-1}$, which performs better under POMDP setting. For both kernels, the $s_t$ and $r_t$ can be generated from the latent prediction $h_t$.

## 3 ENVIRONMENT DYNAMICS DECOMPOSITION

### 3.1 MOTIVATION

An accurate world model is critical in MBRL policy deriving. To decrease the model error, existing works propose various techniques as introduced in Section 1. However, these techniques improve the environment modeling in a black-box manner, which ignores the inner properties of environment dynamics, resulting in inaccurate world model construction and poor policy performance. To address this problem, we propose two important environment properties when modeling an environment:

1) Decomposability: The environment dynamics can be decomposed into multiple sub-dynamics in various ways and the decomposed sub-dynamics can be combined to reconstruct the entire dynamics.

2) Traceability: The environment dynamics can be traced to the action's impact on the environment, and each sub-dynamics can be traced to the impact caused by a part of the action.

For example in the Cheetah task, Figure 1 (b) demonstrates the decomposability: we can decompose the dynamics into $\{thigh, shin, foot\}$ sub-dynamics or $\{back, front\}$ sub-dynamics, which depends on the different decomposition perspectives and the combination of decomposed sub-dynamics can constitute the entire dynamics. Figure 1 (c) explains the traceability: each sub-dynamics can be traced to the corresponding subset of action dimensions: for the *thigh* dynamics, it can be regarded as the impact caused by the *front-thigh* and *back-thigh* action dimensions. The above two properties are closely related to environment modeling: the decomposability reveals the existence of sub-dynamics, which allows us to model the dynamics separately, while the traceability investigates the causes of the dynamics and guides us to decompose the dynamics at its root (i.e. the action).

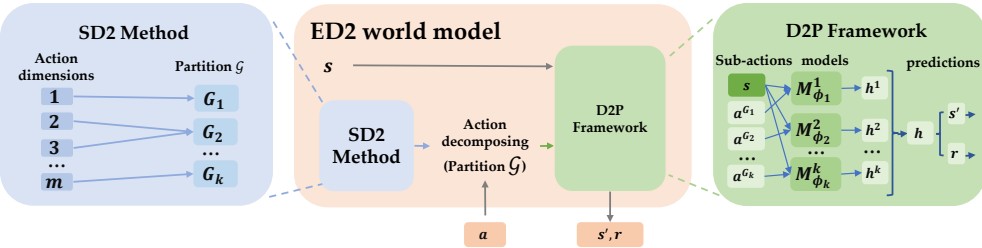

Figure 2: Overview of the world model under ED2 Framework. ED2 contains two components: SD2 and D2P. SD2 decomposes the dynamics by generating partition $\mathcal{G}$ on action dimensions. D2P decomposes action $a$ into multiple sub-actions according to $\mathcal{G}$ and makes decomposing predictions based on $s$ and each sub-action. The prediction $h$ is the combined output of all sub-dynamics models, from which the next state $s'$ and reward $r$ are generated.

To take the above properties into account, we propose the environment dynamics decomposition (ED2) framework (as shown in Figure 2), which contains two key components: sub-dynamics discovery (SD2) and dynamics decomposition prediction (D2P). More specifically, by considering the traceability, we propose to discover the latent sub-dynamics by analyzing the action (SD2, the blue part in Figure 2); by considering the decomposability, we propose to construct the world model in a decomposing manner (D2P, the green part in Figure 2). Our framework can be used as a backbone in MBRL and the combination can lead to performance improvements over existing MBRL algorithms.

## 3.2 DYNAMICS DECOMPOSITION PREDICTION

Given an environment with $m$-dimensional action space $A \subset \mathbb{R}^m$, the index of each action dimension constitutes a set $\Lambda = \{1, 2, \cdots, m\}$, any disjoint partition $\mathcal{G} = \{G_1, \ldots, G_k\}$ over $\Lambda$ corresponds to a particular way of decomposing action space. For each action dimension $i$ in $\Lambda$, we define the action space as $A^i$, which satisfied $A = A^1 \times \cdots \times A^m$. The action space decomposition under partition $\mathcal{G}$ is defined as $A^{\mathcal{G}} = \{A^{G_1}, \cdots, A^{G_k}\}$, where sub-action space $A^{G_j} = \prod_{x \in G_j} A^x$. Based on above definitions, we define the dynamics decomposition for $P$ under partition $\mathcal{G}$ as follows:

**Definition 1** *Given a partition $\mathcal{G}$, the decomposition for $P : S \times A \to S$ can be defined as:*

$$P(s, a) = f_c \left( \frac{1}{k} \sum_{i=1}^{k} P_i(s, a^{G_i}) \right), \forall s, a \in S \times A, \tag{1}$$

*with a set of sub-dynamics functions $\{P_1, ..., P_k\}$ that $P_i : S \times A^{G_i} \to H$, and a decoding function $f_c : H \to S$. Note $H$ is a latent space and $a^{G_i} \in A^{G_i}$ is a sub-action (projection) of action $a$.*

Intuitively, the choice of partition $\mathcal{G}$ is significant to the rationality of dynamics decomposition, which should be reasonably derived from the environments. In this section, we mainly focus on dynamics modeling, and we will introduce how to derive the partition $\mathcal{G}$ by using SD2 in section 3.3.

To implement D2P, we use model $M_{\phi_i}^i$ parameterized by $\phi_i$ (i.e., neural network parameters) to approximate each sub-dynamics $P_i$. As illustrated in Figure 2, given a partition $\mathcal{G}$, an action $a$ is divided into multiple sub-actions $\{a^{G_1}, \cdots, a^{G_k}\}$, each model $M_{\phi_i}^i$ takes state $s$ and the sub-action $a^{G_i}$ as input and output a latent prediction $h^i \in H$. The separate latent predictions $\{h^1, \cdots, h^k\}$ are aggregated and then decoded for the generation of state $s'$ and reward $r$. For each kernel described in Section 2.2, we provide the formal description here when combine with D2P:

$$h_t = \begin{cases} \frac{1}{k} \sum_{i=1}^{k} f(s_{t-1}, a_{t-1}^{G_i}) & \text{For non-recurrent kernel} \\ \frac{1}{k} \sum_{i=1}^{k} f(h_{t-1}, s_{t-1}, a_{t-1}^{G_i}) & \text{For recurrent kernel} \end{cases}$$

We propose a set of kernels, where each kernel models a specific sub-dynamics with the input of current state $s$, corresponding sub-action $a^{G_i}$ and hidden state $h_{t-1}$ (ignored when applying on non-recurrent kernel). The output of all kernels is averaged to get the final output $h_t$. The prediction of reward $r_t$ and state $s_t$ is generated from the output $h_t$. Specifically, we provide an example when combining with the kernel of Recurrent State-Space Model (RSSM) (Hafner et al., 2019) in Figure 3, which is a representative recurrent kernel-based world model. The original single kernel implemented as GRU are replace by multiple kernels with different action input.

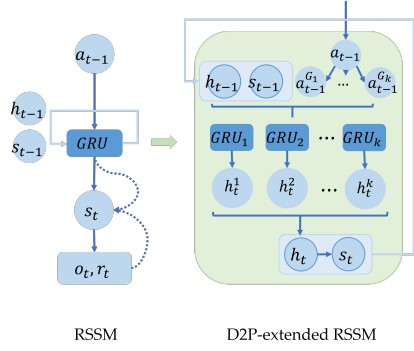

Figure 3: Extension of RSSM with D2P.

### 3.3 SUB-DYNAMICS DISCOVERY

The traceability of the environment introduced in Section 3.1 provides us with a basis for dynamics decomposition: the decomposition on dynamics can be converted to the decomposition on the action space. Therefore, we present the SD2 module for the action space decomposition and discuss three implementations in this section. With the Cheetah task in Section 3.1 as the example: the straightforward SD2 implementation is the complete decomposition, which regards each action dimension as a sub-dynamics and decomposes the dynamics completely. Specifically, complete decomposition decomposes the dynamics into six sub-dynamics: $\{Front, Back\} \times \{Thigh, Shin, Foot\}$. However, complete decomposition ignores the inner action dimensions correlations, which limits its performance in many tasks. For example, the three action dimensions $\{Front\_Thigh, Front\_Shin, Front\_Foot\}$ affect the dynamics of the front part together, thus simply separate these action dimensions would affect the prediction accuracy. To include the action dimension correlations, incorporating human prior for the action space decomposition is an improved implementation. Based on different human prior, we can decompose the dynamics in different ways as introduced in Figure 1. Nevertheless, although human prior considers the action dimension correlations, it is highly subjective and might lead to sub-optimal results due to the limited understanding of tasks (we also provide the corresponding experiment in Section 4.2.3). Therefore, human prior is not applicable in complex systems which is beyond human understanding.

To better discover the sub-dynamics and eliminate the dependence on human prior, we propose to automatically decompose the action space using the clustering-based method. The clustering-based method contains two components: **feature extraction** and **clustering criterion**. Feature extraction extracts the properties of action dimension $a^i$ into feature vector $F^i$. Then we regard each action dimension as a cluster and aggregate related action dimensions together with the clustering criterion. The effectiveness of the clustering-based method depends on the quality of feature extraction and the validity of clustering criteria, which may be different in different environments. Therefore, although we provide a general implementation later, we still suggest readers design suitable clustering-based methods according to task-specific information.

**Feature Extraction:** We extract the properties of each action dimension by computing the Pearson correlation coefficient between action dimensions and state dimensions. Specifically, we define the feature vector as $F^i = \langle |f^{i,1}|, \cdots, |f^{i,n}| \rangle$, where each $f^{i,j}$ denotes the Pearson correlation coefficient between action dimension $i$ and state dimension $j$. $F^i$ describes the impact caused by action dimension $i$ and $f^{i,j}$ is calculated by the corresponding action value $a^i$ and state value changes $\Delta s^j$ (which is the difference between the next state and the current state):

$$f^{i,j} = \frac{cov(a^i, \Delta s^j)}{\sigma_{a^i} \sigma_{\Delta s^j}} \qquad (2)$$

where $cov$ denotes the covariance and $\sigma$ denotes the standard deviation.

**Clustering Criterion:** We define the clustering criterion as the relationship between clusters, which can be formalized as follow:

$$Rela(G_i, G_j) = R(G_i, G_j) - \frac{R(G_j, G_{-i}) \times \omega^{j,-i} + R(G_i, G_{-j}) \times \omega^{i,-j}}{\omega^{i,-j} + \omega^{j,-i}} \qquad (3)$$

where $G_{-i} = \Lambda \setminus G_i$, $\omega^{i,j} = |G_i| \times |G_j|$ and $R(G_i, G_j) = -\frac{1}{\omega^{i,j}} \sum_{A^i \in G_i} \sum_{A^j \in G_j} ||F^i, F^j||_D$. $|| \cdot ||_D$ measures the distance between vectors under distance function $D$ (we choose the negative cosine similarity as $D$).

Algorithm 1 presents the overall implementation of the clustering-based method. As Algorithm 1 describes, with input task $\mathcal{E}$ and clustering threshold $\eta$, we first initialize the cluster set $\mathcal{G}$ containing $m$ clusters (each for a single action dimension), a random policy $\pi_{rand}$, and an empty dataset $\mathcal{D}_c$. Then for $T$ episodes, $\pi_{rand}$ collects samples from the environment and we calculate $F^i$ for each action dimension $i$. After that, for each clustering step, we select the two most relevant clusters from $\mathcal{G}$ and cluster them together. The process ends when there is only one cluster, or when the correlation of the two most correlated clusters is less than the threshold $\eta$. $\eta$ is a hyperparameter which assigned with a value around 0 and empirically adjusted.

---

**Algorithm 1** Selectable clustering-based method.

**Input**: Task $\mathcal{E}$, clustering threshold $\eta$
    Initialize cluster set $\mathcal{G} = \{\{1\}, \cdots, \{m\}\}$ according to $\mathcal{E}$, a random policy $\pi_{rand}$, dataset $\mathcal{D}_c \to \varnothing$
    **for** $i = 1, 2, \cdots, T$ **do**
        Collect and store samples in $\mathcal{D}_c$ with $\pi_{rand}$
    Calculate $F^i$ for each action dimension $i$ with $\mathcal{D}_c$
    **while** $|\mathcal{G}| > 1$ **do**
        $G_{max_1}, G_{max_2} = \arg\max_{G_i, G_j \in \mathcal{G}} Rela(G_i, G_j)$
        **if** $Rela(G_{max_1}, G_{max_2}) > \eta$ **then**
            Remove $G_{max_1}$ and $G_{max_2}$ from $\mathcal{G}$
            Add $G_{max_1} \cup G_{max_2}$ to $\mathcal{G}$
        **else**
            Stop clustering
    **return** $\mathcal{G}$

---

### 3.4 ED2 FOR MBRL ALGORITHMS

ED2 is a general framework and can be combined with any existing MBRL algorithms. Here we provide the practical combination implementation of ED2 with Dreamer(Hafner et al., 2020) (Algorithm 2) and we also combine ED2 with MBPO (Janner et al., 2019) in the appendix. The whole process of ED2-Dreamer contains three phases: 1) SD2 decomposes the environment dynamics of task $\mathcal{E}$, which and can be implemented by three decomposing methods introduced in Section 3.3; 2) D2P models each sub-dynamics separately and constructs the ED2-combined world model $p_\phi$ by

---

**Algorithm 2** ED2-Dreamer

**Input**: Task $\mathcal{E}$, clustering threshold $\eta$
    *// Sub-dynamics Discovery (SD2) Phase:*
    $\mathcal{G} \leftarrow$ SD2 methods $(\mathcal{E}, \eta)$
    *// Dynamics Decomposition Prediction (D2P) Phase:*
    **for** $i = 1, 2, \cdots, |\mathcal{G}|$ **do**
        Build sub-dynamics model: $M_{\phi_i}^i = f(h_{t-1}, s_{t-1}, a_{t-1}^{G_i})$
    Combining all sub-dynamics models: $M_c = \frac{1}{|\mathcal{G}|} \sum_{i=1}^{|\mathcal{G}|} M_{\phi_i}^i$
    Combining $M_c$ with a decoding network $f_{\phi_d}^d$ and construct the ED2-combined world model: $p_\phi = f_{\phi_d}^d(M_c)$
    *// Training Phase:*
    Initialize policy $\pi_\theta$, model: $p_\phi$
    Optimize policy with Dreamer: $\pi_{\hat{\theta}} = \text{Dreamer}(\mathcal{E}, \pi_\theta, p_\phi)$

---

combining all sub-models with a decoding network $f_{\phi_d}^d$; 3) The final training phase initializes the policy $\pi_\theta$ and the world model $p_\phi$, then derive the policy from Dreamer with input $\pi_\theta$, $p_\phi$ and task $\mathcal{E}$.

## 4 EXPERIMENTS

**Baselines & Benchmarks:** For a comparative study, we take two state-of-the-art MBRL methods: MBPO (Janner et al., 2019) and Dreamer (Hafner et al., 2020) as the baselines, and extend them with ED2 as ED2-MBPO, ED2-Dreamer. To reduce implementation bias, we reuse the code and benchmarks from the prior works: the DMC Suite (Tassa et al., 2018) for Dreamer and Gym-Mujoco (Brockman et al., 2016) for MBPO. Besides, we also provide ablation studies to validate the effectiveness of each component of our ED2.

**Clarification:** (1) Although the data required by the clustering-based method is tiny (less than 1% of the policy training), we include it in the figures for a fair comparison. (2) We take the clustering-based method as the main SD2 implementation (denoted as ED2-Methods) and discuss other SD2 methods in Section 4.2.3 and appendix (complete decomposition, human prior are denoted as CD, HP respectively). (3) Due to the space limit, we leave the result of MBPO/ED2-MBPO in the appendix. (4) All results are averaged over 5 seeds. The hyperparameters setting is left in the appendix.

## 4.1 PERFORMANCE

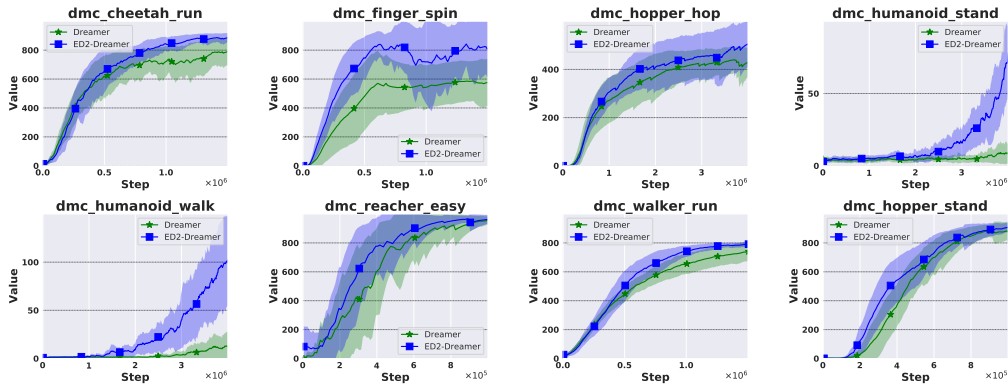

Figure 4: Comparisons of ED2-Dreamer vs. Dreamer. The x- and y-axis represent the training steps and performance. The line and shaded area denotes the mean value and standard deviation.

We evaluate Dreamer and ED2-Dreamer on eight DMC tasks with image inputs. As shown in Figure 4, ED2-Dreamer outperforms Dreamer on all tasks. This is because ED2 establishes a more rational and accurate world model, which leads to more efficient policy training. Another finding is that, in tasks like cheetah_run, and walker_run, ED2-Dreamer achieves lower variance, demonstrating that ED2 can also lead to a more stable training process. Furthermore, Dreamer fails to achieve good performance in difficult tasks such as humanoid_stand and humanoid_walk. In contrast, ED2-Dreamer improves the performance significantly, which indicating the superiority of ED2 in complex tasks. In humanoid tasks, the dynamics are too complex for the clustering-based method with image-based input. Therefore, we use the vector-based state for clustering and keep the policy training on the image-based state (we will further discuss this in Section 5). The performance of MBPO and ED2-MBPO are left in the appendix, which proves that ED2 boost MBPO's performance significantly.

## 4.2 ABLATION STUDIES

### 4.2.1 THE EFFECTIVENESS OF D2P AND SD2

In this section, we investigate the contribution of each component to performance improvement. We can summarize the improvements into three parts: multiple kernels, decomposing prediction, reasonable partition. There is a progressive dependence between these three parts: the decomposing prediction depends on the existence of the multiple kernels and the reasonable partition depends on the decomposing prediction. Therefore, we design an incremental experiment for the validation.

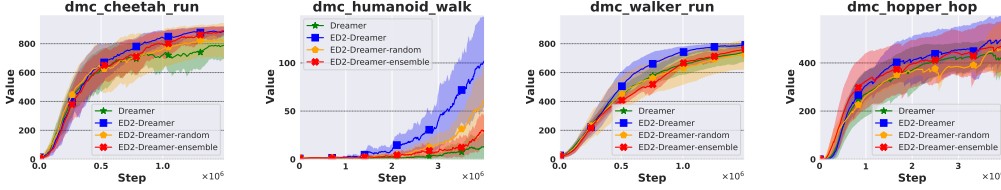

Figure 5: Performance comparisons of components ablation experiments.

First, we employ the ED2-Dreamer-ensemble [2], which maintains the multiple kernel structure but without dynamics decomposing (i.e. all kernels input with action $a$ rather than sub-action $a^{G_i}$). We investigate the contribution of multiple kernels by comparing ED2-Dreamer-ensemble with baselines. Second, we employ the ED2-Dreamer-random, which maintains the D2P structure and obtains partition randomly. We investigate the contribution of decomposing prediction by comparing ED2-Dreamer-random with ED2-Dreamer-ensemble. Last, we investigate the contribution of reasonable partition by comparing ED2-Dreamer with ED2-Dreamer-random.

---

[2]The ensemble refers to kernel ensemble, which is described in detail in the appendix.

Figure 5 shows that ED2-Dreamer-ensemble outperform Dreamer on humanoid_walk and cheetah_run tasks, indicating that multiple kernels help the policy training on some tasks. ED2-Dreamer-random outperforms ED2-Dreamer-ensemble on humanoid_walk task, but not in other tasks (even perform worse in cheetah_run and hopper_hop). This is due to the different modeling difficulty of tasks: the tasks except humanoid_walk are relatively simple and can be modeled without D2P directly (but in a sub-optimal way). The modeling process of these tasks can be aided by a reasonable partition but damaged by a random partition. The humanoid_walk is challenging and cannot be modeled directly, therefore decomposing prediction (D2P) is most critical and performance can be boosted even with a random decomposing prediction. Finally, ED2-Dreamer outperforms ED2-Dreamer-random on all tasks, which indicates that a reasonable partition (SD2) is critical in dynamics modeling and D2P can not contribute significantly to the modeling process without a reasonable partition.

### 4.2.2 MODEL ERROR

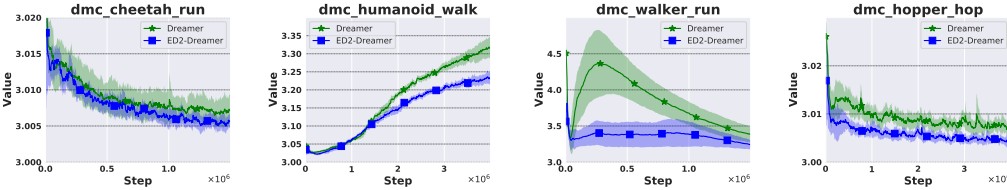

Figure 6: The model error (KL-Divergence) comparison of ED2-Dreamer and Dreamer.

In this section, we further investigate whether the model error is reduced when combined with ED2. We conduct an environment modeling experiment on the same dataset (which is collected in the MBRL training process) and record the model error. Since the policy keeps update in the MBRL training process, the dataset of MBRL also changes with the updated policy. For example, in the MBRL training on the Cheetah task, the model is first trained with the data like *Rolling on the ground* and finally trained with the data like *running with high speed*. To simulate the MBRL training process, we implement our dataset by slowly expanding it from 0 to all according to the data generation time. This setting can also help to investigate the generalization ability of the model on unfamiliar data (i.e. the data generated by the updated policy). We list parts of the result in Figure 6 and the result of other tasks are shown in the appendix.

Figure 6 shows that ED2-Dreamer has a significantly lower model error in all tasks compared with Dreamer. ED2-Dreamer can also achieve a more stable world model training (i.e. with low variance) on humanoid_walk and walker_run tasks. We also find that the baseline methods have significantly increasing model error on humanoid_walk and walker_run tasks, but for ED2-methods, the increase is much smaller. We hypothesize that ED2 produces a reasonable network structure; as the dataset grows, ED2-methods can generalize to the new data better. This property is significant in MBRL since the stable and accurate model prediction is critical for policy training. We also provide the model error comparison of MBPO and ED2-MBPO in the appendix, which also proves that ED2 can reduce the model errors when combine with MBRL methods.

### 4.2.3 SD2 COMPARISON

In this section, we compare the performance of three proposed SD2 methods. We list the decomposition obtained by the clustering-based method and human prior on humanoid_walk task for the illustrating purpose and provide the corresponding performance comparison results in Figure 7. More experimental results on other tasks are provided in the appendix.

As shown in Figure 7, human prior can generate different partitions from different task understandings and we average their performance as the final result. Experiment shows that all SD2 methods help the policy learning. The clustering-based method performs best and baseline Dreamer performs worst in the comparison. For the complete decomposition, it performs poorly under humanoid_walk, which implies that humanoid_walk contains many inner action dimension correlations, and simply complete decomposition heavily breaks this correlation thus hinders the final performance. Compared to complete decomposition, human prior maintains more action dimension correlations by leveraging the human prior knowledge, which leads to better performance. However, the correlations maintained

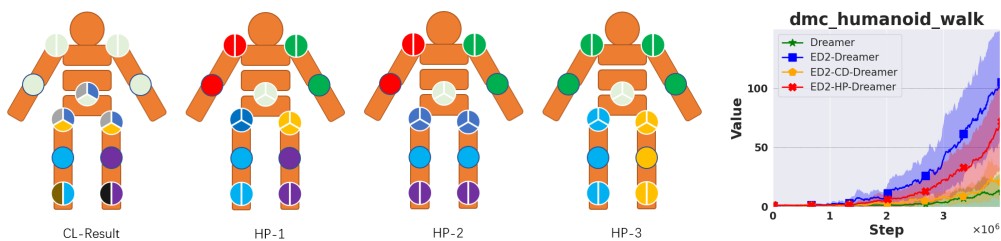

Figure 7: The performance comparison of Dreamer, ED2-Dreamer, ED2-CD-Dreamer and ED2-HP-Dreamer. We provide the sub-dynamics visualization in the left four figures. Each circle correspond to a joint. A joint contains multiple action dimensions when the corresponding circle is separated into multiple parts. We mark the action dimensions in the same sub-dynamics with the same color.

by human prior might be false or incomplete due to human limited understanding of tasks. Compare to human prior, the clustering-based method automatically decomposes the action space according to the clustering criterion, which decomposes the action space better in a mathematical way. For example, human prior aggregates $\{right\_hip\_x, right\_hip\_y, right\_hip\_z\}$ ($x, y, z$ denote the rotation direction) together and the clustering-based method aggregates $\{abdomen\_x, right\_hip\_x, left\_hip\_x\}$ together. Although the action dimensions from human prior sub-dynamics affect the same joint $left\_hip$, they rotate in different directions and play a different role in the dynamics. In contrast, the sub-dynamics discovered by the clustering-based method aggregate the action dimensions that affect the x-direction rotation together. It maintains stronger correlations and helps the world model fitting the movement on x-direction better. Therefore, it performs better than human prior on this task.

## 5 DISCUSSION

In this paper, we regard SD2 as a flexible module that can adopt any suitable partition methods considering the task-specific information. Currently, we discuss three kinds of SD2 methods, i.e., human prior, complete decomposition, and the clustering-based method, and the clustering-based method is chosen as our main implementation since it outperforms the other two methods on these testbeds empirically. We also analyze the reasonable decomposition provided by the clustering-based method, which contributes a lot to the dynamics modeling process. Nevertheless, the clustering-based method is still faced with extra challenges when solving complex tasks with image-based inputs, such as humanoid tasks. In this paper, we use the vector-based state in the clustering stage for humanoid tasks by considering the one-to-one correspondence between vector-based and image-based state representations. How to leverage self-supervised learning or contrastive learning to learn low-dimension, high-quality state features from raw images to improve the partition discovery effect of SD2 and further apply our ED2 to more complex scenarios is worthwhile to further investigate.

Previous work (Doya et al., 2002) also takes the dynamics decomposed prediction into consideration, which achieves better dynamics modeling. It decomposes the dynamics from the perspective of state and time. Different from this work, we analyze the cause of dynamics and decompose it from its root: the action space, which makes the modeling of environmental dynamics more reasonable and scalable.

## 6 CONCLUSION

In this paper, we propose a novel world model construction framework: Environment Dynamics Decomposition (ED2), which explicitly considers the properties of environment dynamics and models the dynamics in a decomposing manner. ED2 contains two components: SD2 and D2P. SD2 decomposes the environment dynamics into several sub-dynamics according to the dynamics-action relation. D2P constructs a decomposing prediction model according to the result of SD2. With combining ED2, the performance of existing MBRL algorithms is significantly boosted. Currently, this work only considers the decomposition on the dimension level, and for future work, it is worthwhile investigating how to decompose environment dynamics at the object level, which can further improve the interpretability and generalizability of ED2.

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

## A  EXPERIMENT SETTING

We keep the experiment setting the same with Dreamer and MBPO. For Dreamer, we do the experiment on Deep Mind Control environments, with images as input, and each episode length is set to 1000. For MBPO, the experiment is processed on the Gym-Mujoco environment, takes the structured data as input, and sets the episode length to 1000. Each experiment is averaged by running five seeds. Our specific computing infrastructure is as shown in Table 1:

Table 1: The computing infrastructure of our experiment.

| CPU | GPU | MEMORY |
|---|---|---|
| XEON(R) SILVER 4214 | RTX2080TI | 256G |

## B  EXPERIMENT HYPERPARAMETERS

For ED2-Dreamer and ED2-MBPO, we followed the official implementation of Dreamer and MBPO except for the dynamics models. Specifically, for the dynamics models, we select the hyperparameters in each environment, as shown in Table 2.

Table 2: The hidden size and $\eta$ value for each environment. DeepMind denotes the environment that belongs to DeepMind Control Suite, and Gym-Mujoco denotes the environment is from Gym-Mujoco.

| ENVIRONMENT | HIDDEN SIZE | $\eta$ |
|---|---|---|
| HOPPER(DEEPMIND) | 200 | 0 |
| WALKER(DEEPMIND) | 200 | -0.06 |
| CHEETAH(DEEPMIND) | 200 | -0.1 |
| HUMANOID(DEEPMIND) | 200 | 0 |
| REACHER(DEEPMIND) | 200 | 0 |
| FINGER(DEEPMIND) | 200 | 0 |
| HALFCHEETAH(GYM-MUJOCO) | 150 | 0 |
| HOPPER(GYM-MUJOCO) | 200 | -0.3 |
| WALKER(GYM-MUJOCO) | 200 | -0.2 |
| ANT(GYM-MUJOCO) | 150 | -0.12 |

For the clustering process, hyperparameter $\eta$ describes the tightness of constraints on inter-group distance and intra-group distance. When $\eta = 0$, the clustering process stop condition is that the distance between the two most relative clusters is equal to the distance between these two clusters and others (this is a general condition in the most environment). In some environments, although $\eta = 0$ is a good choice, but the value can be further finetuned to obtain more reasonable clustering results. The $\eta$ value we use is as in Table 2.

## C  ED2-MBPO IMPLEMENTATION

The combination of MBPO and ED2 can be described as Algorithm 3. We can also separate it into three parts: the first phase is SD2, which discover the sub-dynamics (partition $\mathcal{G}$) in the environment by using appropriate SD2 method. Then the D2P phase models each sub-dynamics separately and construct the ED2-combined world model $p_\phi$. Finally, we derive the trained policy $\pi_{\hat{\theta}}$ by using MBPO method with input task $\mathcal{E}$, initialized policy $\pi_\theta$ and world model ensemble $P_{\hat{\phi}}$.

---

**Algorithm 3** ED2-MBPO

---

**Input**: Task $\mathcal{E}$, clustering threshold $\eta$
    *// Sub-dynamics Discovery (SD2) Phase:*
    $\mathcal{G} \leftarrow$ SD2 methods $(\mathcal{E}, \eta)$
    *// Dynamics Decomposition Prediction (D2P) Phase:*
    **for** $i = 1, 2, \cdots, |\mathcal{G}|$ **do**
        Construct sub-dynamics model: $M_{\phi_i}^i = f(s_{t-1}, a_{t-1}^{G_i})$
    Combining all sub-dynamics models: $M_c = \frac{1}{|\mathcal{G}|} \sum_{i=1}^{|\mathcal{G}|} M_{\phi_i}^i$
    Combining $M_c$ with a decoding network $f_{\phi_d}^d$ and construct the ED2-combined world model: $p_\phi = f_{\phi_d}^d(M_c)$
    *// Training Phase:*
    Initialize policy $\pi_\theta$, model ensemble: $P_{\hat{\phi}} = \{p_{\phi_1}^1, \cdots, p_{\phi_e}^e\}$
    Optimize policy with MBPO: $\pi_{\hat{\theta}} = \text{MBPO}(\mathcal{E}, \pi_\theta, P_{\hat{\phi}})$

---

# D KERNEL ENSEMBLE AND MODEL ENSEMBLE

Kernel ensemble is deployed in section 4.2. We retain the multi kernel network structure and all kernel input with the same information: hidden state $h$ (if combined with recurrent kernel), current state $s$ and current action $a$. Kernel ensemble can be regarded as a single model, which average the outputs from all kernels and training in an end-to-end manner for all kernels.

Model ensemble is widely used in MBRL for uncertainty estimation. In MBRL, the model ensemble is generally implemented by setting different initial parameters and sampling different training data from the same dataset. Models in model ensemble are trained separately and no connection between them except training from the same dataset.

Therefore, model ensemble is totally different from kernel ensemble. Model ensemble propose to train multiple unrelated models for the same task, which can estimate the uncertainty of prediction. But kernel ensemble propose to use one model (but construct with multiple kernels) for the prediction task. We can also combine kernel ensemble with model ensemble and improve the accuracy of all models (e.g. MBPO).

# E MODEL ERROR

In order to verify whether we get a more accurate world model, we measure the model error. Figure 8 is the model error curve of Dreamer / ED2-Dreamer. We can see that our framework can significantly reduce the model error.

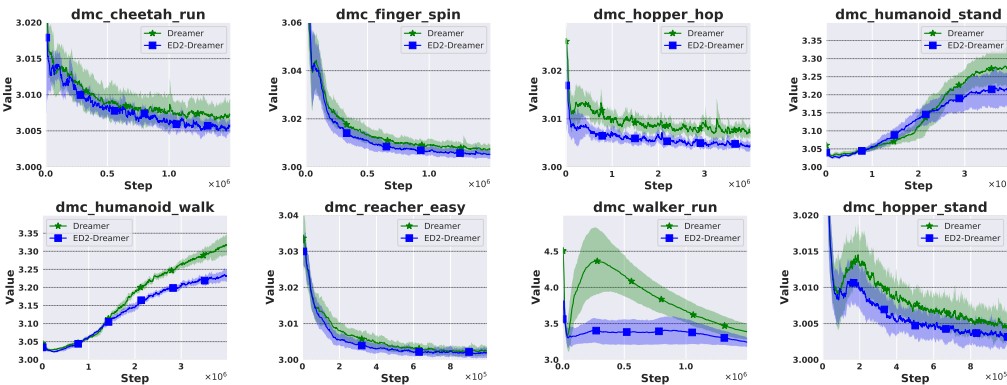

Figure 8: The model error reduced when combine ED2 with Dreamer.

# F ED2-CD-DREAMER

Here we provide the experiment result of complete decomposition in Figure 9. complete decomposition can boost the performance in most environments. But in some complex environments like humanoid and walker, it fails to improve the performance. We analysis that in humanoid and walker, the correlation between action dimensions can't be ignored. Complete decomposition break the correlations between action dimensions and lead to poor performance in these tasks.

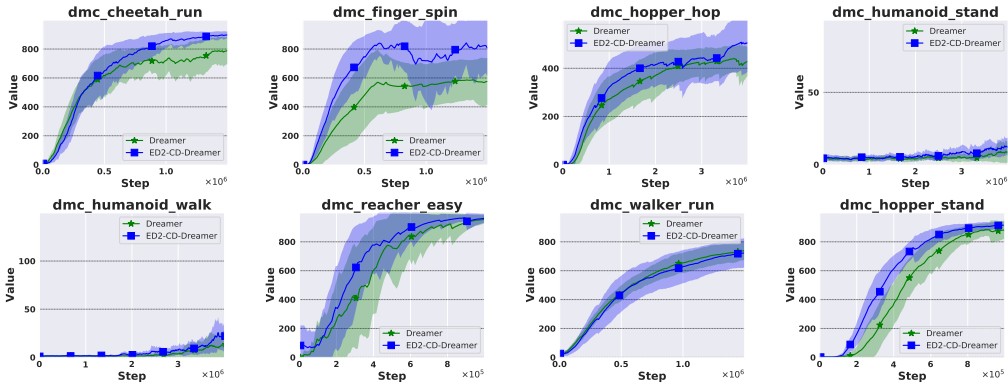

Figure 9: Comparisons between ED2-CD-Dreamer and Dreamer.

# G COMBINE WITH MBPO

Here we provide the experiment results of ED2-MBPO method, which include the performance (Figure 10), model error evaluation (Figure 11) and performance under complete decomposition (Figure 12).

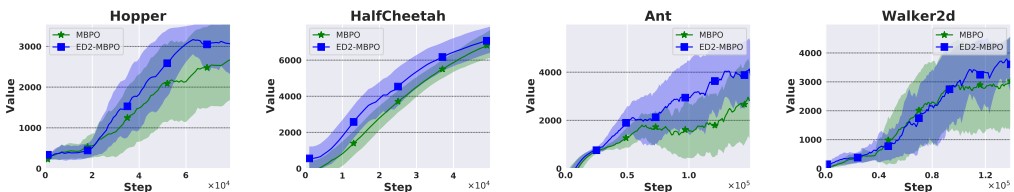

Figure 10: Performance comparisons between ED2-MBPO and MBPO.

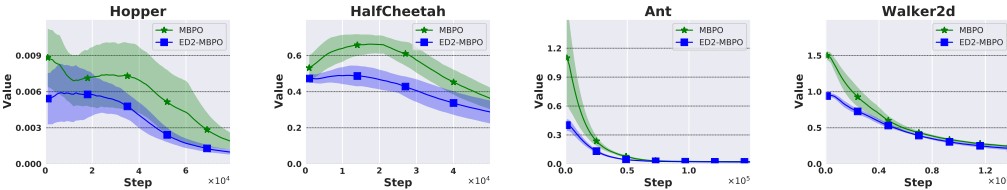

Figure 11: Model error comparisons between ED2-MBPO and MBPO.

# H SD2 CLUSTERING RESULTS

We visualize the final partition result obtained by clustering here with both figure and table form, the figure result of DeepMind Control Suite is shown in Figure 13, and the figure result of Gym-Mujoco is shown in Figure 14. The result shows that the clustering-based method tends to group the relative action dimensions together, and the clustering results are also reasonable from the human point of view.

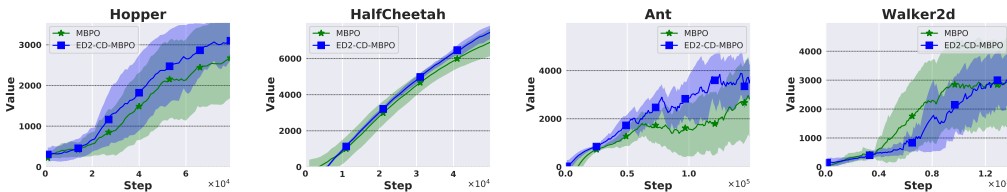

Figure 12: Performance comparisons between ED2-CD-MBPO and MBPO.

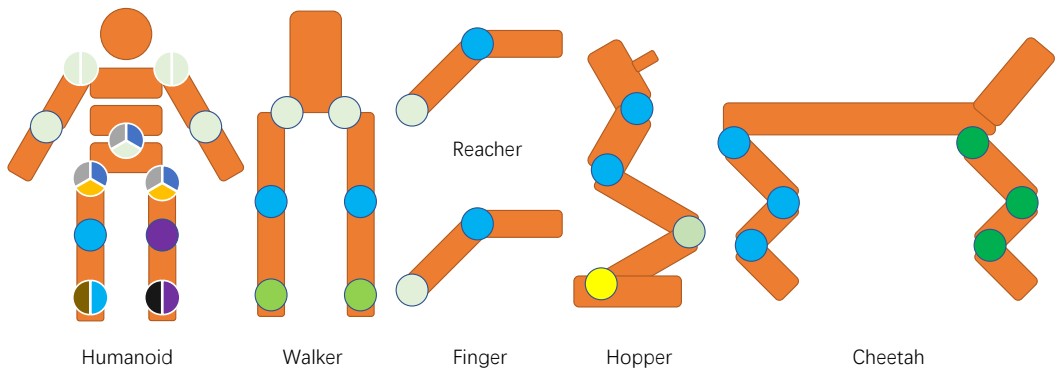

Figure 13: The visualization of final partition of environments in DeepMind Control Suite. Some joints in humanoid is divided into two or three parts (e.g. abdomen joint). It indicates that there are multiple action dimensions contained by this joint (e.g. abdomen joint contains abdomen_x, abdomen_y and abdomen_z action dimensions).

As shown in Table 3, each row denotes a sub-dynamics discovered by the clustering-based method in this environment (expect the final two-row in humanoid environment, they belong to the same sub-dynamics. Because of the length of the table, we write it as two lines). The sub-dynamics we discovered is very reasonable, and there is an obvious connection between the action dimensions in the same sub-dynamics.

## I   ATARI EXPERIMENTS

We also conducted model error experiments on Atari environment and Atari-like Maze environment (called Minecraft) (Omidshafiei et al., 2018). In this experiment, random policy is used to generate data for dynamics model training. As shown in Figure 15, ED2 could bring a more accurate dynamics modeling process.

## J   DREAMER WITH BIGGER HIDDEN SIZE

In this section, we provide the result of Dreamer method under bigger hidden size (which keeps the similar parameter size as ED2-Dreamer). As shown in Figure 16, increasing the size of parameters can not improve the performance.

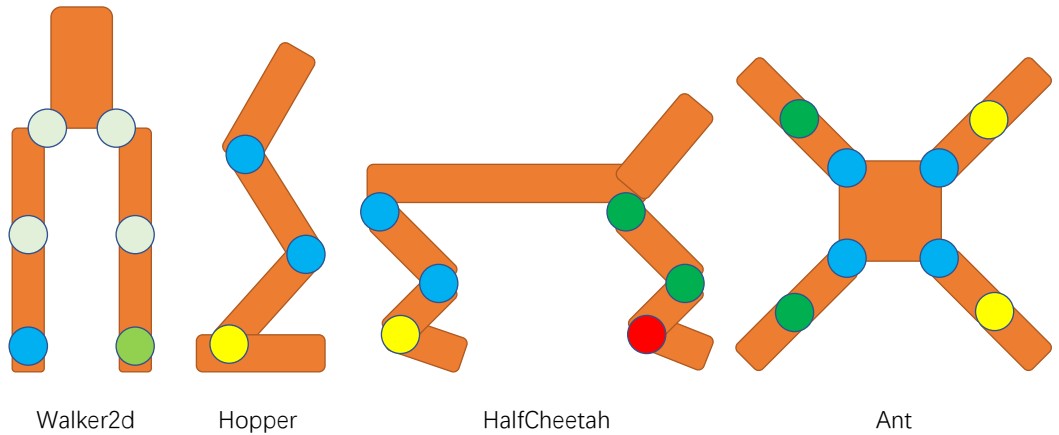

Figure 14: The visualization of final partition of environments in Gym-Mujoco.

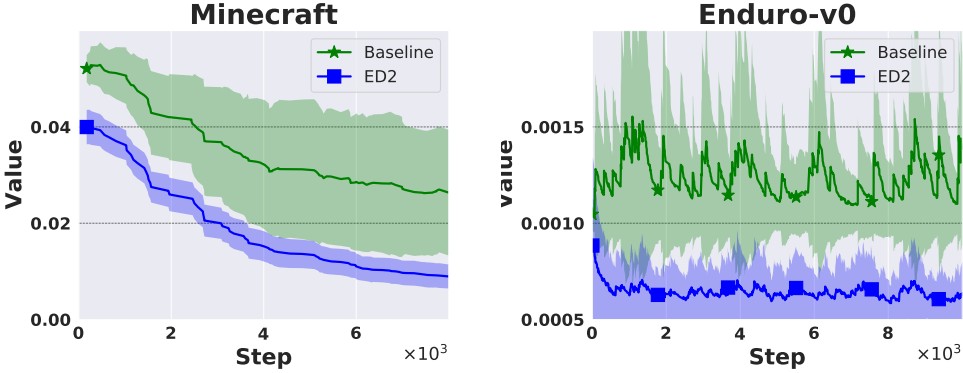

Figure 15: Model error experiments on Atari and Atari like maze environment.

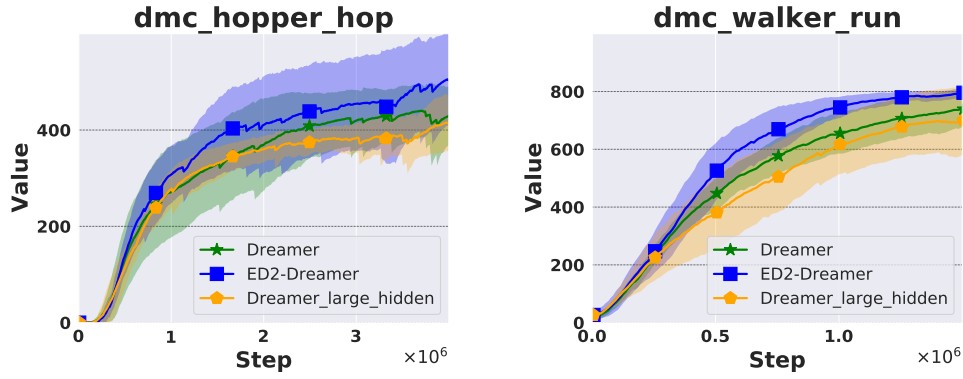

Figure 16: The performance of Dreamer under bigger parameter size.

Table 3: The meaning of action dimensions in each environment (listed according the clustering result)

| ENVIRONMENT | ACTION DIMENSION MEANNING |
|---|---|
| HUAMNOID(DEEPMIND) | RIGHT_ANKLE_X
LEFT_ANKLE_X
ABDOMEN_X, RIGHT_HIP_X, LEFT_HIP_X
ABDOMEN_Y, RIGHT_HIP_Y, LEFT_HIP_Y
RIGHT_KNEE,RIGHT_ANKLE_Y
LEFT_KNEE,LEFT_ANKLE_Y
RIGHT_HIP_Z, LEFT_HIP_Z
LEFT_SHOULDER1, LEFT_ELBOW, RIGHT_SHOULDER1, RIGHT_ELBOW,
RIGHT_SHOULDER2, LEFT_SHOULDER2, ABDOMEN_Z |
| WALKER(DEEPMIND) | LEFT_HIP, RIGHT_HIP
LEFT_KNEE, RIGHT_KNEE
LEFT_ANKLE, RIGHT_ANKLE |
| CHEETAH(DEEPMIND) | BACK_THIGH, BACK_SHIN, BACK_FOOT
FRONT_THIGH, FRONT_SHIN, FRONT_FOOT |
| HOPPER(DEEPMIND) | WAIST, HIP
KNEE
ANKLE |
| REACHER(DEEPMIND) | SHOULDER
WRIST |
| FINGER(DEEPMIND) | PROXIMAL
DISTAL |
| HALFCHEETAH(GYM-MUJOCO) | BACK_THIGH, BACK_SHIN
FRONT_THIGH, FRONT_FSHIN
BACK_FOOT
FRONT_FOOT |
| WALKER2D(GYM-MUJOCO) | RIGHT_THIGH, RIGHT_LEG, LEFT_THIGH, LEFT_LEG
RIGHT_FOOT
LEFT_FOOT |
| HOPPER(GYM-MUJOCO) | THIGH, LEG
FOOT |
| ANT(GYM-MUJOCO) | LEFT_FRONT_HIP, RIGHT_FRONT_HIP, LEFT_BACK_HIP, RIGHT_BACK_HIP
LEFT_FRONT_ANKLE, LEFT_BACK_ANKLE
RIGHT_FRONT_ANKLE, RIGHT_BACK_ANKLE |

