# OpenReview forum: "ED2: An Environment Dynamics Decomposition Framework for World Model Construction"
_ICLR.cc/2022/Conference — ICLR 2022 Submitted_

### Official Review · Reviewer_6o3p · 2021-10-28

**Correctness:** 3
**Technical Novelty And Significance:** 2
**Empirical Novelty And Significance:** 3
**Recommendation:** 5
**Confidence:** 5

**Main Review:**

However, I have the following concerns:

Q1. The sub-dynamics model is to output an embedding vector regarding sub-actions and state $s$, but not to predict the next state $s’$ and reward $r$. The sub-dynamics model is therefore not a dynamics model. I don’t think it is precise to claim that ED2 is a dynamics decomposition method.

Q2. As the underlying idea of ED2 is very simple, I believe it is possible to theoretically analyze the advantage of ED2, which can improve the paper’s contribution significantly.

Q3. Doya et al. [1] also leverage the idea of dynamics decomposition to model the environmental dynamics but decompose the dynamics into multiple domains in state space and time. Although their method is different from ED2, I suspect that it is important to discuss the difference between the two studies because both are based on the idea of dynamics decomposition.

Reference:

[1] Doya, Kenji, et al. "Multiple model-based reinforcement learning." Neural computation 14.6 (2002): 1347-1369.

Q4. The introduction section mentions many MBRL algorithms, such as ME-TRPO [1] and SimPLe [2], but the authors empirically compare ED2 with only two MBRL methods in the experiment section.

References:

[1] Kurutach, Thanard, et al. "Model-ensemble trust-region policy optimization." ICML. 2018.

[2] Kaiser, Łukasz, et al. "Model Based Reinforcement Learning for Atari." ICLR. 2019.


**Summary Of The Paper:**

The authors propose an environment dynamics decomposition (ED2) framework to decompose the environmental dynamics into multiple sub-dynamics. Empirical results show that ED2 improves the performance of several model-based reinforcement learning (MBRL) algorithms. The paper is well-written and easy to read, and the motivation is clear.

**Summary Of The Review:**

Although the paper is well written, the current contribution is not enough for me to recommend acceptance.

---

> ### Author Response · Authors · 2021-11-19
> **To Reviewer 3**
>
> Thank you for your suggestions and review of our paper. We will take your questions seriously and further improve our paper. For the questions you mentioned in the review, we will separate them into the following points and provide our responses.
>
> --output an embedding vector
>
> In MBRL, it is a typical dynamics modeling manner that first predict the latent state and then decode the latent state to get the state and reward. In addition, no matter how well we decompose the action space, there are always a few ignored action dimension relations. Therefore, we hope that after the combination of latent space, the subsequent decoding network can help us further fit the relationship between those action dimensions that we have not considered. Therefore, this implementation is very necessary.
>
>
> -- Multiple model-based reinforcement learning
>
> Thank you for providing the paper, we will mention this paper in the new version, and analyze the relationship between our method and it. This will further help readers understand our method.
>
>
> -- ME-TRPO and SimPLe
>
> These two paper are very old baseline compared to Dreamer and MBPO, so we chose the newer methods for comparison. The Simple method focuses on solving atari problems, which is different from our experimental settings. The reason why we did not conduct experiments on Atari was also given in the reply to Reviewer 1. (Although SIMPLE is published late, it released its work on arXiv long time ago before them publish on ICLR). MBPO and Dreamer are two of the most popular and effective MBRL algorithms, so the comparison with them is very convincing.

---

> > ### Comment · Reviewer_6o3p · 2021-11-25
> > **Response**
> >
> > Thanks for the response that addresses a part of my concerns. I agree that it is reasonable to combine the latent embeddings to predict rewards and successor states. However, I still think ED2 is to decompose the latent space rather than the dynamics, thus it is not very precise to claim that ED2 is a dynamics decomposition method.

---

> > > ### Author Response · Authors · 2021-11-29
> > > **Response**
> > >
> > > Thank you for your reply. We are decomposing the dynamics in latent space, rather than decomposing the latent space simply. The decomposition of dynamics executed on latent space, which will not change the fact that we are decomposing the dynamics.
> > >
> > > For detailed description, we first analyze the relation between dynamics and action space, which reveals the fact that the dynamics is caused by the action. Then we decompose the action space for the dynamics decomposition. Our goal has always been to decompose the dynamics of the environment.
> > >
> > > For the latent space, it is only the algorithm implementation, which is followed by the previous works [1][2]. Although the decomposition is taken in latent space, it will not change the fact that the dynamics are decomposed by our method.
> > >
> > > We would really appreciate it if the reviewer can let us know if they have any pending concerns and if our response addressed some/all of their concerns. We will try to address them before the discussion period ends.
> > >
> > > [1] Hafner D , Lillicrap T , Ba J , et al. Dream to Control: Learning Behaviors by Latent Imagination[J]. 2019.
> > >
> > > [2] Janner M , Fu J , Zhang M , et al. When to Trust Your Model: Model-Based Policy Optimization[J]. 2019.

---

### Official Review · Reviewer_BTNs · 2021-11-02

**Correctness:** 2
**Technical Novelty And Significance:** 2
**Empirical Novelty And Significance:** 3
**Recommendation:** 5
**Confidence:** 3

**Main Review:**

I have some issues with the technical approach, raising doubts over how this method will generalize. In particular in light of the small to moderate improvements in the experiments sections and small selection of benchmarks.

Technical approach:
- The authors make a strong and seemingly very limiting assumption in that they only partition actions, all sub-models get fed the same state input. Then they average the output of all submodels to get the new state, also a strong assumption that the state can be additively decomposed this way. I assume this means their submodels output the full state vector as well, almost like an ensemble approach for the latent state with an extra clustering step to filter out just the action inputs for each sub-network.

- The clustering approach itself also seems to make a very strong assumption in clustering bon linear correlations, and of the action dimensions vs. the state dimensions. I am not sure why this makes sense because linear correlations strongly depend on how you encode the state and action space. For example, it is signed so if you just negate one action dimension, shouldn't that change which other actions it gets clustered with it, even though it will be functionally the same? Further, as dynamics models, shouldn't actions be clustered based on how they predict the next state, not how they are correlated with the current state? In general, the Feature Extraction and Cluster Criterion sections are crucial to this paper but the theoretical basis of the design choices made there are not clear to me.

Experiments:
- The Dreamer and MBPO papers you compare against have 20 and 6 benchmarks respectively, it is unfortunate that you only present a subset of 4 from each. This becomes relevant because the improvements aren't large to begin with and some questionable assumptions were seemingly made in the method section. As you allude to in Appendix F, the method also appears to only improve results on problems that are easily decomposable, which seems reasonable.

Language:
- Readable but some bad grammar. I recommend another pass. In particular, there is frequent overuse of "the", as in "by combining the optimal control algorithm" when you mean "by combining [with] optical control algorithms"

Minor:
- Is it necessary to describe it as a POMDP? The paper by Hafner et al 2020 that this paper builds on does not. Your state space definition is seemingly much more complex than theirs, but as I understand it you use the same benchmarks.

**Summary Of The Paper:**

The authors propose a clustering method to decompose world models for model-based RL into sub-models reflecting the dynamics of the task. The method appears to give small to moderate improvements on most of the presented 4+4 benchmarks.


**Summary Of The Review:**

The problem of learning decomposed world models is interesting and the authors do show a small improvement on most of the presented benchmarks.

Unfortunately, the clustering method, which is the main contribution of the paper, makes a number of strong and unexpected assumptions that make me question how well this approach will generalize. The paper also suffers from poor clarity in the method section on the theoretical motivation for these design decisions. The empirical results are not very strong to begin with, and were only presented for a smaller subset of the benchmarks in the papers they compare against.

---

> ### Author Response · Authors · 2021-11-19
> **To Reviewer 2**
>
> Thank you for your review and suggestions on our paper, which will have an important impact on our paper and help us to further improve this paper.
> As for the concerns you mentioned, we can divide them into the following points:
>
> --network structure
>
> Our approach is relatively simple to implement but provides positive performance gains for multiple MBRL methods. Although the implementation method is similar to Ensemble, we apply the idea of decomposition modeling of different sub-dynamics to achieve better dynamics modeling. The reason why our method performs better than ensemble/baseline is that we consider the correlation of action dimensions and dynamics are decomposed into different sub-dynamics. In the modeling process, the learning of these sub-dynamics will not interfere with each other, and the gradient in the fitting process will not affect each other, so a faster and more accurate modeling process can be achieved. Therefore, although our method implementation is similar to Ensemble, the idea and the way of improvement are completely different.
>
> --clustering method
>
> First, we did not encode the actions and states. We did the clustering algorithm in the original space. Second, we did not calculate the relationship between the action and the current state, we calculate the relationship between the action value and the state change, which describes the impact of the action on the environment. We will clarify this in the revised version to make it clearer.
>
> --Experiments
>
> For Dreamer and MBPO methods, we compare with them on 8 and 4 environments. Our environment accounts for half of the sum of the two papers and provides sufficient experimental results.
>
> --POMDP
>
> In the section 2 of the Dreamer paper, the definition of POMDP is introduced. ED2-Dreamer has the same environment as Dreamer, so we need to introduce the definition of PODMP

---

> > ### Comment · Reviewer_BTNs · 2021-11-24
> > **Response to Rebuttal**
> >
> > I thank the authors for their candid responses. Clustering based on correlation with changes in state makes more sense, but I'm still concerned that clustering actions based on simple linear correlation is dependent on the action encoding and may not generalize well. For example, flipping the sign of an action in the dynamics function seems like it would change the clustering result? The clustering approach seems like a central weakness of the approach, and you even admit in the paper that one may need application-specific clustering.
> >
> > Overall I think it's an interesting approach, and the discussion you gave to another reviewer about how you ended up using this type of architecture where you take the mean of the "sub-dynamics" networks would have been very helpful to have (and empirically evaluated) in the paper itself. As it stands, I unfortunately find it difficult to raise my score.

---

> > > ### Author Response · Authors · 2021-11-29
> > > **Response**
> > >
> > > Thank you for your reply. There might exist some misunderstandings about our paper. Our clustering method will not change the final clustering effect with the action value changes.  Our clustering method is based on a large amount of data, therefore, we are analyzing the action-state relations under many different action values. Therefore, the action value we select will not affect the final clustering result.
> > >
> > > Based on the above introduction, we would like to clarify the application-specific clustering. In our opinion, no algorithm that can solve all problems with a given calculation process and parameters (for example, DQN is difficult to solve the problem of continuous space). And we want to clarify that SD2 and D2P are separate parts, and the implementation of SD2 is not limited in our paper (although our approach is already generalizing). If there is a better sub-dynamic approach to a practical problem than the SD2 approach we provide, then we recommend readers to use it for dynamics decomposing and then combine it with the D2P we provide.
> > >
> > > As a summary, we would like to clarify that our contributions are from two parts: D2P and SD2. D2P is the main framework we provide, and SD2 is a relatively open method, and we will not limit readers to the algorithm we provide. We hope readers can choose the appropriate dynamic decomposition method to decompose the action space, and then apply the idea of dynamic decomposition prediction to complete the modeling of the environment, and finally improve the effect of the MBRL method.

---

> > > > ### Comment · Reviewer_BTNs · 2021-11-29
> > > > **Reviewer Response.**
> > > >
> > > > You are correct that it was a bad example, as you are clustering based on linear correlations in a learned latent space. However, my point seems to stand that clustering on linear correlations is rather limited?
> > > >
> > > > First, this is effectively approximating a learned (non-linear) NN with another univariate linear model (instead of e.g. ARD or computing sensitivities of this NN). Second, wouldn't clustering on the absolute value of correlations have made more sense? E.g. unless I'm still missing something, action dimensions that end up negatively correlated with state changes (in this latent space) will not cluster with dimensions that were positively correlated, even though they may both strongly affect the same state. This seems rather simplistic compared to many other methods proposed on model structure learning. While this paper may be the first to apply the concept to a Dreamer latent space, since the "Dynamics Decomposition" in the title is a core contribution, I would really have liked to see more investigation and motivation of your choices here.

---

> > > > > ### Author Response · Authors · 2021-11-30
> > > > > **Response to Reviewer 2**
> > > > >
> > > > > Thank you for your timely reply, which made us clearly understand the misunderstanding between us.
> > > > >
> > > > > We need to clarify that we do use the absolute value for clustering, which corresponds to Section 3.3, Feature extraction, Line 3. We cluster the action dimensions according to the action-state correlations, without considering the positive correlation or negative correlation, which is consistent with your opinion. As long as two action dimensions are influencing similar states, we will take them into consideration together. I think this is the main misunderstanding between us.
> > > > >
> > > > > Also in our paper, we provide three implementation examples of SD2, and we only declare that these are strong generalization methods, but it does not mean that they can be perfectly applied in all environments. For some corner cases, our methods may not be directly applied, so we suggest readers choose according to the actual situation. For SD2 problems, on the one hand, they can be solved more easily according to the actual situation of the problem; on the other hand, the academic problems are very big and need more future work to further solve them. Our contribution is to come up with this idea of decomposition prediction, not a perfect solution to all environments.
> > > > >
> > > > > Thank you again for your reply. If you have any other concerns, please contact us and we will try our best to solve these problems.

---

> > > > > > ### Comment · Reviewer_BTNs · 2021-11-30
> > > > > > **Reviewer Response**
> > > > > >
> > > > > > I re-read the updated paper now and the clustering part is clearer, although it could have been even clearer by stating that you are using what seems to be simple agglomerative clustering of action dimensions, based on vectors representing their linear correlations with changes in the state vector. The other limitations of using simple linear correlations for this remains, and ideally I would have liked to see some more analysis of why you chose this clustering approach and how well it generalizes.
> > > > > >
> > > > > > I actually think that decomposing the latent space is an interesting approach, but learning a good model decomposition is generally, as the authors also note, a very difficult problem. The execution and validation of this could have been stronger in this paper. As it is I'm not convinced that this simple clustering approach will generalize outside these examples, and while tailoring the clustering approach to other environments is an option as the author notes, without this the paper is much weaker. The  "decomposition prediction", which I assume refers to the additive way you combine the sub-dynamics models your clustering found, is not thoroughly motivated or analyzed either. If the experiments at least had been more exhaustive, I would have been happy to raise my score despite these apparent weaknesses. As it stands I'm not confident that I can motivate raising my score. However, I hope that this paper, when properly motivated by further benchmarks or theory, will ultimately end up being published somewhere.

---

### Official Review · Reviewer_SSLz · 2021-11-02

**Correctness:** 4
**Technical Novelty And Significance:** 2
**Empirical Novelty And Significance:** 2
**Recommendation:** 3
**Confidence:** 4

**Details Of Ethics Concerns:**

No concerns

**Main Review:**

This paper proposed to build multiple world models, one per action space partition, and average it to improve the model error.
This idea is generic to other MBRL approaches, and the paper implements the method in "Dreamer" and "MBPO" and shows improvements in the model error.

#### Strengths
* The idea is simple, and the experiment results showed improvements in model error or value in some domains

#### Weakness
* If I understood correctly, this action space partitioning can only be applicable when there is a clear decomposable structure in the action space.
* The result shows some marginal improvements on the five random seeds.

#### Questions
* Have you experimented with this dynamics decomposition approach to other domains like ATARI?

* Could you provide a side-by-side comparison of NN on ED2-Dreamer and simple Dreamer?
Will the total number of parameters remain the same or similar? Does the performance changes per the choice of hyperparameters?

* In Figure 6, there is a gap in the model errors between the two approaches. If we run on more sample interactions, will this gap disappear? Or remains the same?

* The main difference is that ED2-Dreamer usually shows lower variance over the five random seeds than Dreamer. Is this due to the averaging effect of the dynamic decomposition? In Figure 5, ED2-Ensemble takes the whole action space, learns multiple models, and averages the results. We can see that the blue and the red curves are less similar in the cheetah and hopper domains. Would it imply that the decomposition doesn't bring improvements in those domains?

* When combining the latent states, the paper proposed to use averaging the outcomes. Have you tested other choices like max or putting an additional layer?




**Summary Of The Paper:**

This paper presents a model-based RL algorithm that provides decomposed dynamics models when the state and action spaces are defined as a multi-dimensional Cartesian space.
We can divide the method into two parts:
* The one for partitioning the action coordinates, and
* The NN architecture, one for each partition and then combining the outcomes ($h^i_t$) of partitions into the final one ($h_t$).
This paper also shows a clustering based on the correlation, human-provided partition, and full/random partitioning algorithms.

The experiment shows the results on Mujoco-based environments that have orthogonal coordinates per position and velocity, so
the action coordinate partitioning can be readily applicable.

**Summary Of The Review:**

This paper shows MBRL methods that utilized the decomposition of the action space and showed experiment results on the control domains.
In some domains, we see improvements in the model error or higher values. But the overall improvement is marginal, and the method can only be applicable to the problems having readily decomposable action space.

---

> ### Author Response · Authors · 2021-11-19
> **To Reviewer 1**
>
> We sincerely appreciate your responsible reviews and constructive suggestions, and we would like you to know that your questions provide helpful guidance to improve the quality of our paper. We can summarize your concerns as follows:
>
> -- only be applicable to the environment with a clear action space decomposition structure.
>
> Our method can be applied to all typical RL environments (e.g., Mujoco, Atari), not just the ones mentioned in this paper. We have considered a number of environments, including Mujoco, Atari-like maze and Atari environments. For example, in the complex humanoid environment, without a clear action space decomposition structure, we can decompose them according to the action-state relations. In another example like Atari-like maze environments, we can decompose its dynamics completely: {go_up, gp_down, gp_right, go_left} with action space {up, down, right, left}, results show that we can construct the dynamics model more accurately. In Atari environment, we can discover the sub-dynamics by using the complete decomposition method and get a more accurate world model (we will provide the results of maze and Atari later in new paper version). Therefore, our method can be well applied in the Mujoco and Atari environments. These two types of environments are currently the most widely used environments, and our method can achieve better results. Although there are still environments we may have omitted, our method performs better in most environments that are widely used as the benchmarks in RL.
>
> The reason we don't present the results on Atari and the maze environment is that it's quite simple to apply ED2 to these environments (not that it can't be applied to these environments). In these environments, different actions cannot affect the environment at the same time (for example, we cannot choose to go-left and go-up at the same time), so there is no correlation between the different actions, we just need to completely decompose the action space. The Mujoco environment is the more challenging environment for ED2. However, this makes a misunderstanding on generalization of ED2, so we will also provide the results on the Atari environments in the new version.
>
>
> --The improvement is marginal.
>
> Firstly, our method improves marginally in a small number of environments and greatly in most environments, especially in difficult Humanoid environments. Our proposed approach is a general modeling framework, so it may have different enhancement effects in different environments. The effect of our approach on MBRL training has two parts: positive and negative. The positive part is that we decompose the dynamics, so the complex environment dynamics fitting task will be decomposed into several sub-tasks to solve, making the modeling task simpler. The negative part is that the dynamic decomposition may break the relationship between certain action dimensions (this problem is mitigated when we group high-related action dimensions together). In some experiments, the positive part brings many benefits and the negative part is almost zero, which provides a huge performance improvement. However, in some environments, the correlation between the action dimensions is strong, which may result in some negative influence, so the improvement is not obvious. However, in all environments, the positive effects of ED2 are higher than the negative effects, but the degrees are different. Therefore, there are some environments where our approach improves marginally. But our approach improves the MBRL approach as a whole, rather than focusing on a single environment.
>
> Second, the biggest contribution of our approach is that we can make all MBRL methods achieve improvement on almost all tasks with a simple modification. The performance gains contributed by our approach are general and consistent.
>
> Third, our approach retains almost the same hyperparameters as the baseline approach and does not use tuning to achieve a larger performance gap. It shows that our method improves the algorithm directly and does not depend on parameter adjustment.
>
> -- ATARI Experiment.
>
> We have done experiments on Atari environments and the Atari-like maze environment, and the results show that decomposition prediction can be used to model environments more accurately. The maze environment is Minecraft, from the paper CASL[1] and the Atari environment is Enduro-v0. We take the image as the state and regard each action dimension as a sub-dynamics. We train the world model with a random policy and evaluate the modeling accuracy. The baseline model is implemented as the world model in the paper SLBO [2]. Results show that ED2 helps world model construction and we will provide the experiment result in the new paper version.

---

> > ### Author Response · Authors · 2021-11-19
> > **The remaining content**
> >
> >
> >
> > -- Comparison of ED2-Dreamer and simple Dreamer
> >
> > We provide these experiments in the paper. For example, ED2 Dreamer and ED2-ensemble Dreamer are comparative experiments with the same number of parameters. The number of kernels in ED2-ensemble Dreamer is always the same as the number of sub-dynamics in ED2-Dreamer, even ED2-ensemble has the same parameters amount, it still underperforms ED2. For the experiments with the same number of parameters, we not only have the ED2-ensemble experiments mentioned in the paper, but also tried to increase the hidden state of the baseline Dreamer algorithm, so that the number of parameters of the baseline Dreamer method is the same as that of ED2, and then compare them with ED2. The results showed that increasing the number of parameters in the baseline method did not help the MBRL training process. Specific experimental results will be presented in the new version.
> >
> > In ED2-Dreamer, we reuse all the hype-parameters of baseline Dreamer and did not finetune them (this can be verified by the code we submitted). We believe that the hyperparameter in baseline Dreamer was the most suitable hyperparameter for them, and we achieved performance improvements without fine-tuning, which showed that our approach can boost the performance directly and is not sensitive to hyperparameters.
> >
> > -- gap in the model errors
> >
> > In some tasks, ED2 has a faster convergence speed, but with enough training time, the baseline can converge to the same result as ED2 (like finger_spin, reacher_easy). However, in most tasks, ED2 can obtain lower model errors than baseline, which can not be changed by the training times (like humanoid tasks, hopper_hop). This difference is due to different environmental complexities. For complex environment, it is difficult to model directly, so decomposition modeling of dynamics can bring better modeling convergence effect. However, for some simple environments, decomposition prediction can make the modeling process faster, but when the amount of training is sufficient, baseline modeling method can also converge to a good result.
> >
> >
> > -- lower variance
> >
> > In environment dynamics modeling, ensemble and averaging operation help some environments get smaller training variances. This is one of the reasons Ensemble is widely used in MBRL. However, in ED2, the smaller variance is not caused the average operation. In ED2, different kernels have different inputs, resulting in different outputs (different sub-dynamics). At this time, average operation is used to fuse information, rather than to stabilize the training process. Therefore, in ED2, the more stable training process comes from the rational decomposition of the environment. In conclusion, both ensemble and reasonable decomposition can bring a more stable training process. However, in ED2, the existence of multiple kernels is not the ensemble implementation, so its stable performance comes from reasonable environmental decomposition.
> >
> > -- combining the latent states
> >
> > Thank you for your advice. During the research process, we tried the information fusion methods of concatenation, max operation, putting some additional layer and attention mechanism, but the final experimental effect shows that the average operation are better than those methods.
> >
> > As a summary, it is important to clarify that our method is not limited to Mujoco environments, but can be combined with all environments we have seen. Even in Atari-type environments, our approach can achieve better results.
> >
> > Reference：
> > [1] Omidshafiei S, Kim D K, Pazis J, et al. Crossmodal Attentive Skill Learner[C]//Proceedings of the 17th International Conference on Autonomous Agents and MultiAgent Systems. 2018: 139-146.
> > [2] Luo Y, Xu H, Li Y, et al. Algorithmic Framework for Model-based Deep Reinforcement Learning with Theoretical Guarantees[C]//International Conference on Learning Representations. 2018.

---

### Decision · Program_Chairs · 2022-01-20

**Decision:**

Reject

**Comment:**

One of the major challenges in model-based RL is the learning of accurate world models. This work proposes a method that can learn to decompose the dynamics of the world into several sub-dynamics, which is postulated to lead to better prediction accuracy (which in turn should lead to better policies/downstream performance). The proposed method clusters the sub-dynamics in latent space, and can be combined with any existing MBRL method. Here it is specifically combined with Dreamer and MBPO and evaluated on the deepmind control suite/mujoco.

**Strengths**
This work addresses an important problem, and on a high-level the problem/approach is well motivated
The proposed algorithm is "simple" (which can be a really good thing) and very general in that it can be combined with seemingly any MBRL method

**Weaknesses**
The manuscript was lacking in clarity on a technical level (partially addressed during rebuttal)
On average the experimental results are not very convincing (yet)

**Rebuttal**
The authors clarified a few misunderstandings the reviewers and also updated the manuscript accordingly

**Summary**
I agree with the reviewers that the experimental results are not fully convincing. When looking at the model error plots, the y-scale is very small, and it looks like there is no significant improvement. Also in the "downstream tasks" only for the humanoid do we seem to see a significant improvement. Overall this seems promising, but the authors should investigate why there seems to be a clearer benefit for the humanoid, and show more such results. Furthermore, there are some concerns/clarity issues with respect to what your approach learns - I would recommend you take a small-ish toy example to introduce the intuition of your approach, and maybe visualize learned sub-dynamics.
Overall, while promising, in it's current state this manuscript is not quite ready yet for publication